# SKM-TEA: A Dataset for Accelerated MRI Reconstruction with Dense Image Labels for Quantitative Clinical Evaluation

**Arjun D. Desai, Andrew M. Schmidt, Elka B. Rubin, Christopher M. Sandino, Marianne S. Black, Valentina Mazzoli, Kathryn J. Stevens, Robert Boutin, Christopher Ré, Garry E. Gold, Brian A. Hargreaves, Akshay S. Chaudhari**
Stanford University
{arjundd,...,akshaysc}@stanford.edu

## Abstract

Magnetic resonance imaging (MRI) is a cornerstone of modern medical imaging. However, long image acquisition times, the need for qualitative expert analysis, and the lack of (and difficulty extracting) quantitative indicators that are sensitive to tissue health have curtailed widespread clinical and research studies. While recent machine learning methods for MRI reconstruction and analysis have shown promise for reducing this burden, these techniques are primarily validated with imperfect image quality metrics, which are discordant with clinically-relevant measures that ultimately hamper clinical deployment and clinician trust. To mitigate this challenge, we present the *Stanford Knee MRI with Multi-Task Evaluation (SKM-TEA)* dataset, a collection of quantitative knee MRI (qMRI) scans that enables end-to-end, clinically-relevant evaluation of MRI reconstruction and analysis tools. This 1.6TB dataset consists of raw-data measurements of ~25,000 slices (155 patients) of anonymized patient MRI scans, the corresponding scanner-generated DICOM images, manual segmentations of four tissues, and bounding box annotations for sixteen clinically relevant pathologies. We provide a framework for using qMRI parameter maps, along with image reconstructions and dense image labels, for measuring the quality of qMRI biomarker estimates extracted from MRI reconstruction, segmentation, and detection techniques. Finally, we use this framework to benchmark state-of-the-art baselines on this dataset. We hope our SKM-TEA dataset and code can enable a broad spectrum of research for modular image reconstruction and image analysis in a clinically informed manner. Dataset access, code, and benchmarks are available at https://github.com/StanfordMIMI/skm-tea.

## 1 Introduction

Magnetic resonance imaging (MRI) is a life-saving and sensitive tool for non-invasively diagnosing neurological, musculoskeletal, oncological, and other abnormalities [47]. However, MRI data acquisition is inherently slow and can last up to an hour per patient, which can limit patient throughput in hospitals and can lead to increased patient wait times. Additionally, while MRI has enabled high-resolution anatomical imaging, identifying and quantifying pathology requires dense annotations (e.g. segmentations), which are cumbersome to curate and are prone to inter-reader variations. These limitations have stifled widespread and timely access to and analysis of MRI exams, leading to delayed or missed diagnoses while increasing the already burgeoning costs of healthcare.

Recently, machine learning (ML) has been utilized to create new algorithms for reconstructing high quality images while only sparsely sampling raw MRI data, which accelerates MRI scans and reduces overall scan time [21]. Similarly, ML-based segmentation and detection tools have shown success in

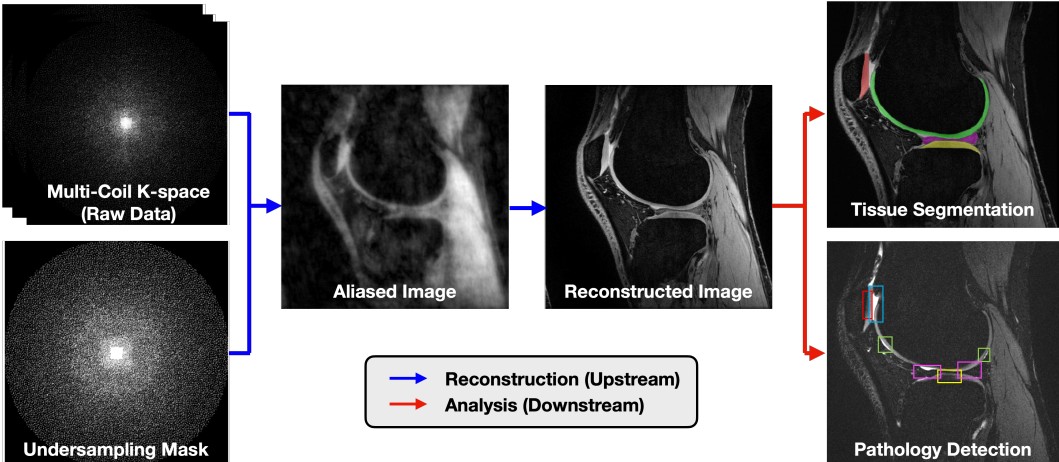

Figure 1: An overview of the end-to-end multi-coil MRI pipeline (and corresponding ML tasks). First, undersampled data acquired by multiple sensor coils is transformed into high quality images (i.e. reconstruction, blue arrow). Then, tissue regions of interest are localized (e.g. segmentation and detection) during image analysis. The SKM-TEA dataset curates raw data, ground-truth images, and dense annotations to enable all tasks. It also offers both a *Raw Data Benchmarking Track*, which supports all these tasks, and the *DICOM Benchmarking* track, which supports all image analysis tasks (red arrow).

automating dense image labeling and in some cases, have even reached performance within the range of inter-reader variability of experts [2].

However, despite the high performance of new ML techniques for MRI reconstruction and analysis on existing benchmarks, few tools have successfully been deployed prospectively in clinics [6]. This translational barrier may be attributed to *metric discordance* - the lack of agreement between true clinical utility of reconstructed MR images and annotations versus popular image quality analyzers (IQAs, e.g. peak signal-to-noise ratio [pSNR] and structural similarity [SSIM]), and pixel-wise or surface-based segmentation metrics (e.g. Dice, intersection over union [IOU], or average symmetric surface distance [ASSD]) [13]. While these metrics provide a standardized and quantitative way of evaluating different techniques, the low sensitivity of these methods to clinically relevant features limit their utility in clinical decision-making. This issue is particularly prevalent in MRI reconstruction, where gold-standard evaluation requires time-consuming and expensive expert readings of images by radiologists. This has led to low concordance between radiologist assessments and ML metrics for determining image quality, and even competition-winning reconstruction models miss important pathologies that require surgical followup [24]. This problem is further exacerbated due to the lack of large-scale public datasets that include both the raw MRI data along with multiple clinically-relevant observations for each image.

We seek to address these challenges with the *Stanford Knee MRI with Multi-Task Evaluation (SKM-TEA)* dataset, a collection of raw MRI data, corresponding images, quantitative biomarkers, and dense tissue and pathology labels that together facilitate clinically relevant evaluation of MRI reconstruction and analysis methods. The main contributions of this work are as follows:

(1) **End-to-end MRI:** We curate a dataset of clinically-acquired quantitative MRI (qMRI) knee scans, images, and dense labels, which together can be used to generate clinically-meaningful, localized tissue-wise qMRI biomarker maps for end-to-end benchmarking of the MRI reconstruction and analysis pipeline.

(2) **Clinically-relevant evaluation:** We propose a standardized framework for evaluating model-generated outputs (reconstructions and dense labels) against these qMRI biomarker maps, which can serve as endpoints for quantitatively measures of tissue health.

(3) **Benchmarking:** We benchmark state-of-the-art MRI reconstruction and segmentation models using both traditional image and label metrics and the new clinically-relevant, quantitative MRI metrics enabled by this dataset.

Table 1: A summary of publicly available knee datasets, supported tasks, their size, and their inputs.

| Dataset | Tasks | | | | Size (>20k slices) | qMRI |
|---|---|---|---|---|---|---|
| | Reconstruction | Classification | Segmentation | Detection | | |
| mridata [20, 36] | ✓ | ✗ | ✗ | ✗ | ✗ | ✗ |
| fastMRI(+) [24, 49] | ✓ | ✓ | ✗ | ✓ | ✓ | ✗ |
| MRNet [2] | ✗ | ✓ | ✗ | ✗ | ✓ | ✗ |
| OAI [40] | ✗ | ✓ | ✓ | ✗ | ✓ | ✓ |
| SKM-TEA | ✓ | ✓ | ✓ | ✓ | ✓ | ✓ |

Dataset access, code, and evolving benchmarks are available at `https://github.com/StanfordMIMI/skm-tea`. Code is also distributed as a Python package: `pip install skm-tea`.

## 2 Background

Here, we provide a brief description of MRI reconstruction problem and introduce quantitative MRI.

**MRI reconstruction:** MRI scans require long acquisition durations because the raw-data that is acquired during imaging is the point-by-point Fourier transform of the desired image, termed as the *k-space*. Sampling this k-space at the Nyquist sampling frequency is necessary to avoid image artifacts (i.e. aliasing). Accelerated MRI is a classical inverse problem consisting of subsampling the acquired raw k-space data below the Nyquist rate (that leads to artifactual images) and subsequently solving the ill-posed problem of recovering the original high-quality diagnostic image [21].

**Quantitative MRI (qMRI):** Most MRI scans are fundamentally qualitative in nature, where clinical diagnoses are made based on *relative* signal intensity differences between different image regions (i.e. are two adjacent tissues lighter/darker than normal?). Newer qMRI methods encode quantifiable physical and biochemical parameters directly into the images to ascribe a physical quantity per pixel, which enables cross-sectional and longitudinal studies of human health [5]. While qMRI methods are gaining popularity, their implementation is hampered due to extremely long scan times and the need to manually localize regions-of-interest (ROIs) in the image [40, 29, 41]. The availability of raw k-space data for qMRI studies will not only assist in defining new and improved local pixel-wise reconstruction accuracy metrics, but also encourage widespread adoption of rapid qMRI methods.

The 3D quantitative double-echo steady-state (qDESS) MRI method allows acquiring rapid qualitative and quantitative knee imaging. qDESS acquires 2 sets of inherently-registered 3D images (termed echoes - E1 and E2) with varying image contrasts that can be used to compute the qMRI parameter of $T_2$ relaxation time (as shown in Fig. 2), which is sensitive to collageneous tissue degeneration [46]. qDESS images have been used as a standalone 5-minute method for diagnostic knee MRI [7, 4] and can extract quantitative biomarkers for osteoarthritis in both knees [3, 19, 26].

## 3 Related Work

Previous datasets for MRI reconstruction, segmentation, and classification have been essential for enabling ML benchmarks for MRI research. However, these datasets are either limited in their size, application scope, or evaluation criteria. Table 1 summarizes the attributes of these datasets.

**Reconstruction:** mridata.org provides fully sampled raw 3D MRI k-space datasets for 19 healthy-subjects in an acquisition that took 40+ minutes each for scan [20, 36]. The small dataset size and homogeneous composition of healthy subjects makes it challenging to analyze how different reconstruction methods affect subtle pathology and clinical outcomes. Additionally, since this sequence is not commonly used in clinics, there exists a fundamental distribution shift between models built for this data acquisition and those that are clinically commonplace [10]. fastMRI is a large repository of k-space data acquired for ≈1600 2D knee and ≈7000 2D brain MRI scans of varying contrasts, with each imaging volume averaging 30-40 slices per acquisition [25]. The extent of images made available through fastMRI and the global challenges has catalyzed ML research for MRI reconstruction [24, 34]. However, as described in the 2019 knee MRI reconstruction challenge, even for some of the challenge winning methods, conventional IQAs did not accurately convey true clinical imaging quality and also clearly obscured pathology [24]. This depicts a clear discordance between metrics used for ML evaluation and those for clinical image interpretation.

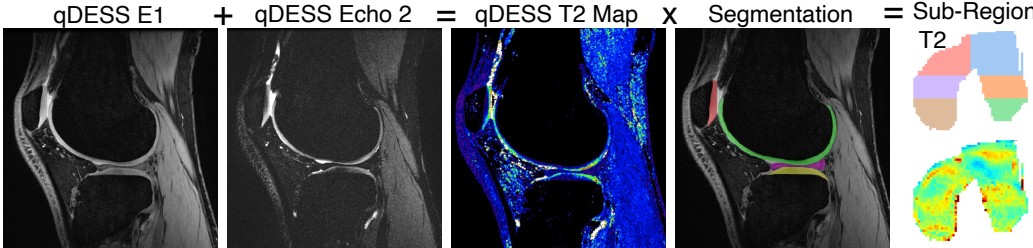

Figure 2: An overview of the qMRI parameter estimation pipeline. Reconstructed qDESS echo images are used to estimate quantitative $T_2$ parameters per pixel (i.e. $T_2$ maps). Tissue segmentations of articular cartilage and the meniscus are used to create tissue-wise $T_2$ maps. These segmentations are also automatically divided into sub-regions to get more localized regions of $T_2$ (example 3D femoral cartilage projected onto 2D). Regional $T_2$ estimates computed using ground truth data are compared to estimates produced using new methods for reconstruction or image segmentation.

**Segmentation:** There are several datasets for dense tissue-level segmentations for MRI data from varying anatomies; however, none of these also include the raw k-space data that is required for MRI reconstruction [27, 8]. Segmentation metrics are often also evaluated with quantitative volumetric or surface-based segmentation measures such as Dice and ASSD, instead of more clinically-relevant features. This often leads to a question regarding "how much performance is good enough?" which cannot be answered without domain-specific clinical knowledge. Without image segmentation, extracting tissue-specific qMRI parameters of interest is also rendered impossible since most clinical use cases focus on specific tissues within an image, rather than the entire image.

**Classification/Detection:** The MRNet and the Osteoarthritis Initiative (OAI) datasets provide labels for classification of knee MRI abnormalities [2, 40]. However, these classifications are only provided at a patient-level, and there are no abnormality bounding boxes to permit localization of the abnormality amongst multiple 3D imaging volumes. These datasets also do not provide the raw k-space, which hampers end-to-end evaluation of the impact of reconstruction methods on downstream clinical utility. Furthermore, the OAI datasets provides labels for the MRI Osteoarthritis Knee Score (MOAKS) on a subset of patients studied, which is similar, but not identical to routine clinical evaluation [23]. The fastMRI+ extension complements scans from the fastMRI reconstruction dataset with detection labels of clinical pathology [49]. While these labels are useful for localizing pathology in reconstructed images and for developing downstream detection models, evaluation still relies on standard IQA and detection metrics, which are discordant with clinically relevant endpoints.

**Overall Need:** Despite the large scale of data that has been made available, a limitation regarding optimal techniques for evaluating accelerated MRI reconstruction, especially for qMRI methods, still remains. Most datasets focus on a single specific evaluation task, which makes it challenging to simultaneously evaluate the benefit of acquisition or analysis methods. Segmentations to drive clinically-useful decisions exist for a handful of imaging datasets, but these have neither qMRI information nor raw k-space data. This motivates datasets that include raw k-space data for building image reconstruction methods with simultaneous evaluation metrics for clinically meaningful outcomes in a mutli-task manner to enable better cross-talk amongst the ML and medical imaging communities.

## 4 Dataset

SKM-TEA consists of raw k-space MRI and image data collected from clinical knee qMRI scans, de-identified patient and imaging metadata, and corresponding dense tissue and pathology annotations for multi-task evaluation as shown in Fig. 1. In this section, we detail the data curation process for the raw data, images, and annotations as well as recommendations for distribution and usage. Additional details can be found in Appendix A.

### 4.1 Data Collection

**Collection overview:** 155 patients at Stanford Healthcare received a knee MRI with the qDESS sequence on one of two 3 Tesla (3T) GE MR750 scanners (GE, Waukesha, WI) with parameters shown in Table 5 (Appendix A.1). Both k-space data and scanner-generated DICOM images were

collected, de-identified and stored securely. All data was collected with patient consent and Stanford University Institutional Review Board approval.

**Raw data (k-space):** All complex raw k-space data (with real and imaginary channels) were acquired in a multi-coil setting with 2x1 parallel imaging with elliptical sampling. Unsampled k-space data was subsequently synthesized using Autocalibrating Reconstruction for Cartesian imaging (ARC) with the GE Orchestra MATLAB SDK (v1.4) and was considered to be the fully-sampled k-space.

**DICOM images:** Standard reconstruction pipelines implemented by MRI vendors generate magnitude images that are distributed in the DICOM format [39, 28]. Since these methods may involve proprietary image and signal filtering, these images are not easy to reproduce. They are not interchangeable with images generated from the raw data and should not be used as a target for reconstruction tasks. However, scanner-generated DICOM images are useful for medical image analysis tasks (classification, segmentation, etc.) and are thus released with this dataset.

**$T_2$ qMRI DICOM parameter maps**: Along with standard volumetric images, we also generated $T_2$ parameteric maps from the qDESS E1 and E2 images in the DICOM format using an analytical signal model described in [46]. Similar to the volumetric images, these underwent vendor-propriety post-processing. For this reason, these maps should only be used for visualization purposes. All analysis that uses $T_2$ maps should use the open-source implementation of this signal model in DOSMA (v0.1.0) [12, 11] to generate $T_2$ maps from the qDESS echoes.

**Tissue segmentations**: Manual segmentation for the following tissues were performed on all DICOM images: (1) patellar cartilage, (2) femoral cartilage, (3, 4) lateral and medial tibial cartilage, and (5, 6) lateral and medial meniscus.

**Localized pathology labels**: The DICOM images from all patients were reviewed alongside their radiology reports that described sixteen pathological categories across joint effusion and meniscal, ligament, and cartilage lesions. These reports were subsequently translated into 3D bounding boxes.

## 4.2 Data Preparation

**Hybridized k-space and SENSE reconstructions:** The 1D orthogonal inverse Fourier transform was applied to the fully-sampled k-space along the readout dimension to generate a hybridized k-space ($x \times k_y \times k_z$). As the readout direction is always fully-sampled, this operation alleviates memory constraints of modern computational accelerators. Sensitivity maps for each 2D ($k_y \times k_z$) slice were estimated using JSENSE (implemented in SigPy [37]) with a kernel-width of 6 and a $24 \times 24$ center k-space auto-calibration region [48]. The fully-sampled k-space was then reconstructed using SENSE [42]. We refer to the images reconstructed from the fully-sampled k-space and estimated sensitivity maps as *SENSE reconstructions* to distinguish them from scanner-generated DICOM images.

Scanner-generated DICOM images undergo proprietary image filtering known as *gradient warping corrections*, which leads to distortions between the SENSE-reconstructed images and the DICOM images (more details in Appendix A.3) [16]. Thus, all analysis or end-to-end inter-operation between SENSE reconstructions and tissue segmentations should use the gradient-warp-corrected segmentation. All image analysis starting only with the DICOM images should use the provided DICOM segmentations.

**Dataset splits:** Of the 155 unique knee MRI scans, data from 36 patients who received additional arthroscopic surgical intervention were included in the test set. The remaining 119 scans were randomly split into 86 and 33 scans for training and validation, respectively. Data from all splits is provided publicly. However, for benchmarking purposes, all models should only be trained using data in the training and validation splits. All metrics should be reported on the test split.

**Distribution:** Raw data, sensitivity maps, and SENSE reconstructions are distributed as pixel arrays in the HDF5 format. Image data and all segmentations are distributed in multiple formats (HDF5, DICOM, NIfTI) to facilitate interoperability between ML and clinical visualization workflows. Patient protected health information (PHI) metadata was anonymized from DICOM files, which were all subsequently manually inspected. Raw data, sensitivity maps, and scanner-generated and SENSE reconstructions were manually checked for PHI and quality. For additional details on distribution and maintenance, see Appendix A.5.

# 5 Dataset Tracks

The SKM-TEA dataset enables two tracks with multi-task evaluations: (1) the Raw Data Benchmark and (2) the DICOM Image Benchmark. In this section, we discuss the available tasks in each track and best practices to ensure reproducibility for future work. Table 6 summarizes the available data and the track(s) with which they are compatible. When distributing or publishing benchmarks on this dataset, please report the specific dataset and `skm-tea` package versions used for all experiments (see Appendix A.6 for usage details).

## 5.1 Raw Data Benchmark Track

The *Raw Data Benchmark Track* relates to all tasks that leverage the raw k-space data and any artifacts generated from or related to the raw data. Data in this track enables tasks pertaining to MRI reconstruction, image segmentation, pathology detection, and qMRI parameter estimation.

**Reconstruction:** This task evaluates the quality of images reconstructed from undersampled (i.e. accelerated) MRI acquisitions. All model reconstructions are evaluated with respect to the complex-valued SENSE parallel-imaging reconstructions, which are state-of-the-art and clinically accepted, using fully-sampled raw data. Because all provided raw data, including that in the test set, is considered fully-sampled, k-space undersampling masks must be generated to simulate accelerated acquisitions. To ensure reproducibility among all test results, fixed undersampling masks are provided for each scan at acceleration factors of $R = 4, 6, 8, 10, 12, 16$. All undersampled inputs at test time should be generated using these fixed undersampling masks at the appropriate acceleration.

**Segmentation and Detection:** Tissue segmentation masks and localized pathology labels and bounding boxes can enable segmentation and detection tasks as well as localized image quality for evaluating reconstruction tasks in clinically pertinent regions. When using these labels in the context of this track, all image inputs correspond to ground-truth SENSE-reconstructed images or reconstructions generated directly from the raw data without additional post-processing. Gradient-warp corrected segmentations should be used as the segmentation masks in this track.

**Multi-Task MRI:** Given spatially-localized image labels are aligned with SENSE reconstruction targets, data in this track can enable multi-task learning setups that capture the end-to-end imaging workflow, from reconstruction to analysis (segmentation, detection, and qMRI generation).

## 5.2 DICOM Benchmark Track

The *DICOM Benchmark Track* enables image analysis tasks using scanner-generated, magnitude DICOM images and corresponding tissue segmentations and detection labels. While data in this track is not compatible with reconstruction tasks, DICOM images have historically been the standard for downstream image analysis tasks like segmentation, detection, and classification [38, 27, 13].

**Segmentation and Detection**: Like in the previous benchmarking track, tissue and pathology labels can enable benchmarking of segmentation, detection, and other image analysis models. Segmentations without gradient-warp correction should be used for this track.

# 6 Evaluation and Benchmarks

In this section, we introduce a standardized evaluation pipeline for using quantitative parameter map estimates and dense annotations to quantify clinically-relevant indicators as a new metric for performance of reconstruction and segmentation models. We use this metric, along with standard image quality and pixel-wise metrics, to benchmark models for a subset of tasks in each track.

## 6.1 $T_2$-based qMRI evaluation

qMRI workflows to ascertain the biochemical status of tissues are highly sensitive to two key intermediate steps prior to retrieving clinically-relevant quantitative metrics: (1) image reconstructions must be high quality to allow for good parametric map estimation; and (2) regions of interest (ROIs) must be precisely segmented (or otherwise localized) to ensure per-pixel qMRI parameters are measured over the correct region. Due to their ability to measure disease endpoints and their sensitivity

Table 2: Peak-signal-to-noise ratio (pSNR) and structural similarity (SSIM) [mean (standard deviation)] for SKM-TEA reconstruction baselines for echoes E1 and E2 accelerated (Acc.) at 6x and 8x.

| | Metric | pSNR (dB) | | SSIM | |
|---|---|---|---|---|---|
| Acc. | Model | E1 | E2 | E1 | E2 |
| 6x | U-Net (E1/E2) | 31.5 (1.38) | 33.7 (1.02) | 0.77 (0.027) | 0.73 (0.032) |
| | U-Net (E1+E2) | 31.1 (1.38) | 33.2 (1.05) | 0.77 (0.024) | 0.74 (0.030) |
| | U-Net (E1⊕E2) | 31.1 (1.63) | 33.5 (1.02) | 0.76 (0.026) | 0.73 (0.034) |
| | Unrolled (E1/E2) | **35.0 (1.08)** | 34.5 (1.09) | 0.83 (0.024) | 0.76 (0.031) |
| | Unrolled (E1+E2) | 35.0 (1.07) | **34.5 (1.09)** | **0.84 (0.022)** | **0.76 (0.030)** |
| | Unrolled (E1⊕E2) | 35.0 (1.08) | 34.2 (1.08) | 0.83 (0.023) | 0.76 (0.030) |
| 8x | U-Net (E1/E2) | 30.6 (1.55) | 32.9 (1.02) | 0.73 (0.030) | 0.67 (0.035) |
| | U-Net (E1+E2) | 30.8 (1.24) | 32.5 (1.00) | 0.72 (0.030) | 0.68 (0.035) |
| | U-Net (E1⊕E2) | 30.8 (1.23) | 32.7 (1.03) | 0.72 (0.029) | 0.68 (0.039) |
| | Unrolled (E1/E2) | 33.8 (1.07) | 33.7 (1.06) | 0.79 (0.027) | **0.73 (0.033)** |
| | Unrolled (E1+E2) | 33.8 (1.07) | 33.6 (1.07) | **0.80 (0.027)** | 0.71 (0.035) |
| | Unrolled (E1⊕E2) | **33.9 (1.08)** | **33.9 (1.07)** | 0.80 (0.027) | 0.73 (0.033) |

Table 3: Performance [mean (standard deviation)] of SKM-TEA reconstruction baselines with respect to absolute $T_2$ error (in milliseconds) for articular cartilage and the meniscus localized with ground truth segmentations. Typical cartilage $T_2$ values are 30-40ms, while meniscus $T_2$ values are 10-15ms).

| Acc. | Model | Patellar Cartilage | Femoral Cartilage | Tibial Cartilage | Meniscus |
|---|---|---|---|---|---|
| 6x | U-Net (E1/E2) | 2.19 (1.68) | 1.08 (0.94) | 1.61 (0.95) | 2.70 (1.35) |
| | U-Net (E1+E2) | 2.83 (1.95) | 2.46 (1.88) | 1.46 (0.92) | 2.01 (1.42) |
| | U-Net (E1⊕E2) | 1.77 (1.50) | 1.11 (0.78) | 1.54 (1.03) | 1.81 (0.97) |
| | Unrolled (E1/E2) | **0.563 (0.23)** | **0.765 (0.28)** | **1.03 (0.42)** | 2.48 (0.79) |
| | Unrolled (E1+E2) | 0.570 (0.23) | 0.836 (0.32) | 1.12 (0.42) | 2.52 (0.78) |
| | Unrolled (E1⊕E2) | 1.69 (1.36) | 2.01 (0.92) | 1.34 (0.55) | **1.31 (0.82)** |
| 8x | U-Net (E1/E2) | 3.48 (1.74) | 2.71 (1.37) | 3.21 (1.24) | 3.76 (1.10) |
| | U-Net (E1+E2) | 2.66 (2.06) | 3.04 (2.03) | 1.49 (1.16) | **2.39 (1.31)** |
| | U-Net (E1⊕E2) | 1.29 (1.09) | 1.26 (0.91) | 2.09 (1.13) | 2.49 (1.80) |
| | Unrolled (E1/E2) | **0.721 (0.30)** | **0.899 (0.34)** | **1.26 (0.49)** | 2.78 (0.87) |
| | Unrolled (E1+E2) | 0.971 (0.42) | 0.988 (0.39) | 1.30 (0.49) | 2.86 (0.88) |
| | Unrolled (E1⊕E2) | 0.588 (0.29) | 0.992 (0.43) | 1.33 (0.63) | 2.73 (0.89) |

to these steps [6], qMRI pipelines may be relevant targets for evaluating MRI reconstruction and image analysis methods.

We propose a qMRI-based evaluation framework that uses the $T_2$ parameter maps generated from the qDESS scans as a target for regional quantitative parameter analysis, as shown in Fig. 2. In this pipeline, reconstructions of the two qDESS echoes are used to generate a $T_2$ parameter map. Tissue segmentations are subsequently applied to mask relevant ROIs in the parameter map to localize tissue-specific qMRI values. These ROIs can be further divided into physiologically-relevant sub-regions for each tissue. In the knee, the articular cartilage and meniscus, along with their sub-regions, have shown high predictive power for the early-onset of chronic disease [1]. To extract the sub-regional parameters, tissue segmentations are automatically subdivided into these sub-regions, and $T_2$ parameters are averaged over these sub-regions (more sub-regional details in Appendix B). To ensure reproducibility, all $T_2$ parameter map estimation and tissue sub-region division is done with the open-source implementation in DOSMA (v0.1.0).

New reconstruction and image analysis methods can be used to generate image or label inputs into this pipeline. Differences in estimated $T_2$ values obtained from ground truth and predicted reconstructions and labels can be used to measure the accuracy of these estimates. Details on the pipeline used for each track can be found in Appendix C.

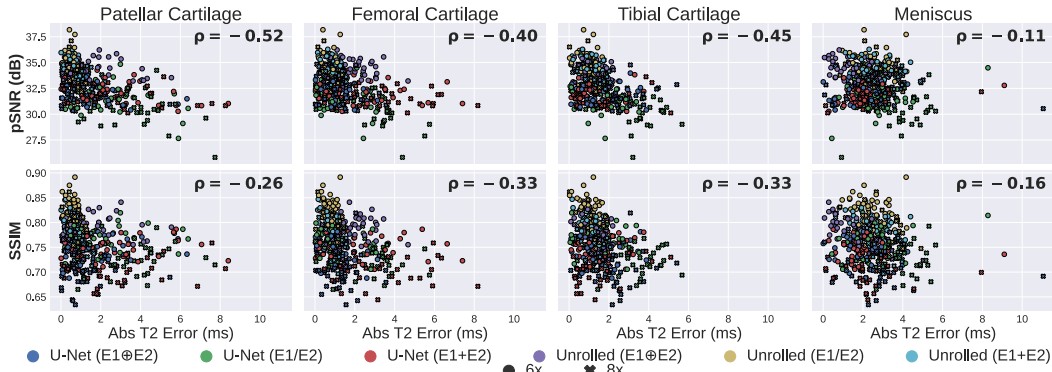

Figure 3: Performance among various Raw Data Track reconstruction models (colored markers) on image quality metrics versus absolute $T_2$ estimation error at both 6x (circle) and 8x (cross) acceleration. Top – peak-signal-to-noise ratio (pSNR), bottom – structural similarity (SSIM). Both pSNR and SSIM were very weakly correlated ($|\rho| \leq 0.52$) with $T_2$ estimation errors across all tissues.

## 6.2 Baselines

We benchmark and summarize popular state-of-the-art models for MR reconstruction and segmentation on SKM-TEA below (more details on training setup in Appendix D).

**Raw Data Track – Reconstruction:** 2D U-Net [43] and unrolled [15, 44] networks were trained to reconstruct 2D undersampled, complex-valued axial slices at 6x and 8x acceleration. As each scan consists of two 3D images (E1 and E2 echoes) for reconstruction, models were trained with the following configurations: (1) two separate models for echoes 1 and 2 (E1/E2); (2) a single model for both echoes, with each echo a unique training example (E1+E2); or (3) a single model for both echoes, with echoes 1 and 2 as multiple channels in a single example (E1⊕E2).

**DICOM Track – Segmentation:** 2D V-Net [30] and U-Net models were trained on DICOM images to segment patellar cartilage, femoral cartilage, tibial cartilage, and the meniscus. Separate models were trained on: (1) echo 1 only (E1), (2) echo 2 only (E2), (3) multi-channel echo1-echo2 (E1⊕E2), and (4) the root-sum-of-squares (RSS) of the two echos.

## 6.3 Metrics

Standard image quality and segmentation metrics, in addition to the proposed $T_2$ evaluation framework, were used to evaluate reconstruction and segmentation models, respectively. Image reconstruction performance was measured using peak-signal-to-noise ratio (pSNR) and structural similarity (SSIM) on both qDESS echoes. Segmentation performance was measured with dice similarity coefficient (DSC), average symmetric surface distance in millimeters (ASSD, mm), volumetric overlap error (VOE), and coefficient of variation (CV). Both $T_2$ error ($T_2^{pred} - T_2^{gt}$) and absolute $T_2$ error ($|T_2^{pred} - T_2^{gt}|$) (and their bias/variance) were used to evaluate estimated $T_2$ precision and accuracy.

## 6.4 Results and Analysis

**Raw Data Track – Reconstruction:** Among reconstruction models, unrolled models outperformed U-Net models in both pSNR and SSIM at both accelerations and across both echoes (Table 2). Unlike standard fully convolutional networks (e.g. U-Net), which rely on statistical imaging priors learned during training, unrolled networks can encode prior information about the image formation model with proximal update steps, which can improve image recovery. In addition, among models of the same architecture, there was a negligible performance difference in how multi-echo inputs were input into the models. While standard IQA metrics indicated reasonably high performance, $T_2$ error performance of these models was more variable (Table 3). U-Net (E1+E2) had the least bias for patellar and tibial cartilage $T_2$ estimates at 6x and for patellar cartilage at 8x. U-Net approaches had higher variance (>1.0 ms) in these estimates compared to unrolled models. Models trained and evaluated on 6x-accelerated scans predominantly had lower variance, but higher bias in $T_2$ estimates. However, Unrolled (E1⊕E2) had higher accuracy and lower variance in $T_2$ estimates of patellar

Table 4: V-Net segmentation performance measured by standard ML pixel and surface segmentation metrics with absolute $T_2$ error. Models were trained with echo 1 only (E1), echo 2 only (E2), multi-channel echo 1 and echo 2 (E1$\oplus$E2), and the root-sum-of-squares (RSS) of both echoes.

| Metric | Tissue | E1 | E2 | E1$\oplus$E2 | RSS |
|---|---|---|---|---|---|
| DSC | Patellar Cartilage | 0.88 (0.082) | 0.85 (0.11) | **0.89 (0.084)** | 0.88 (0.088) |
| | Femoral Cartilage | 0.88 (0.035) | 0.86 (0.033) | **0.88 (0.029)** | 0.88 (0.033) |
| | Tibial Cartilage | 0.86 (0.036) | 0.83 (0.048) | 0.86 (0.034) | **0.86 (0.034)** |
| | Meniscus | 0.85 (0.059) | 0.83 (0.052) | **0.85 (0.056)** | 0.85 (0.060) |
| ASSD (mm) | Patellar Cartilage | **0.33 (0.28)** | 0.49 (0.63) | 0.36 (0.64) | 0.36 (0.54) |
| | Femoral Cartilage | 0.26 (0.11) | 0.29 (0.088) | **0.25 (0.081)** | 0.25 (0.096) |
| | Tibial Cartilage | 0.33 (0.11) | 0.41 (0.17) | 0.32 (0.10) | **0.32 (0.089)** |
| | Meniscus | **0.49 (0.27)** | 0.54 (0.23) | 0.49 (0.29) | 0.54 (0.50) |
| Abs T2 Error (ms) | Patellar Cartilage | **0.64 (0.47)** | 1.02 (0.76) | 0.80 (0.71) | 0.66 (0.48) |
| | Femoral Cartilage | 0.53 (0.38) | 0.87 (0.51) | **0.49 (0.35)** | 0.55 (0.40) |
| | Tibial Cartilage | 0.49 (0.51) | 0.87 (0.76) | **0.47 (0.51)** | 0.52 (0.52) |
| | Meniscus | **0.52 (0.63)** | 0.91 (1.0) | 0.71 (0.82) | 0.58 (0.66) |

cartilage at 8x-acceleration compared to 6x-acceleration. Given the higher signal in articular cartilage in E1 compared to E2, joint optimization of multi-channel (E1$\oplus$E2) inputs may result in implicit over-prioritization of reconstruction of the E1. This may lead to more variable per-pixel recovery for E2. As a result, per-pixel T2 maps, which are computed based on pixel-wise ratios between E2/E1, may have more variability and error. At higher acceleration factors, networks optimized with the l1-loss tend to blur the image (i.e. average the signal). Signal averaging may reduce the per-pixel variance and lead to more accurate region-wide signal, and thus T2, estimates.

**DICOM Track – Segmentation:** Among all segmentation approaches, E1, $E1 \oplus E2$, and RSS models had the highest and similar performance. E2 had consistently worse performance on both pixel-wise metrics and $T_2$ accuracy measures, likely due to the worse E2 echo image contrast. Additionally, low SNR of E2 compared to E1 may result in higher variability in segmentations, and thus more variability in $T_2$ accuracy. Other models had similar average performance and variance across DSC, VOE, and CV metrics. All approaches overestimated $T_2$ in patellar cartilage, femoral cartilage, and tibial cartilage, but underestimated $T_2$ in the meniscus. These methods also had higher variance (>0.6ms), which may indicate higher variability in the estimates despite low bias. V-Net models achieved higher performance compared to U-Net models among standard ML segmentation metrics (DSC – patellar cartilage, ASSD – all tissues), but had similar performance among $T_2$ error metrics. The discordance between relative differences for ML and qMRI metrics may suggest that standard ML metrics may not be a wholistic representation of relative performance differences between models. Thus, the use of only ML metrics for benchmark comparison may lead to inconsistencies between reported performance and practical utility. Results from U-Net benchmarks and additional metrics are detailed in Appendix E.1.

**Metric Concordance:** To understand concordance between standard ML metrics and $T_2$ accuracy, we quantified the Pearson correlation coefficient ($\rho$) between image quality and pixel-wise metrics with absolute $T_2$ estimation error across all models. Reconstruction metrics had very weak correlation with $T_2$ estimation error across all four tissues (both $|p| \leq 0.52$). A similar trend was observed for sub-regions of each of these tissues (Appendix E.3). Both segmentation metrics (DSC and ASSD) were also very weakly correlated (both $|p| \leq 0.4$) with absolute $T_2$ error (Fig. 4). Thus, using clinically-relevant $T_2$ biomarkers as direct endpoints for quantifying performance can help mitigate the challenge of low concordance between standard ML metrics and accuracy of $T_2$ estimates.

# 7  Limitations and Ethical Considerations

First, all data was acquired from patients who received a knee MRI scan at Stanford Hospital, which is a specific subgroup of the general population based on geographic, demographic, and health insurance constraints. Second, although data was collected from two different scanners, both scanners were from the same vendor, which may affect model performance on data acquired from different vendors. Given the heterogeneity of patient anatomy, and MRI acquisition and processing techniques,

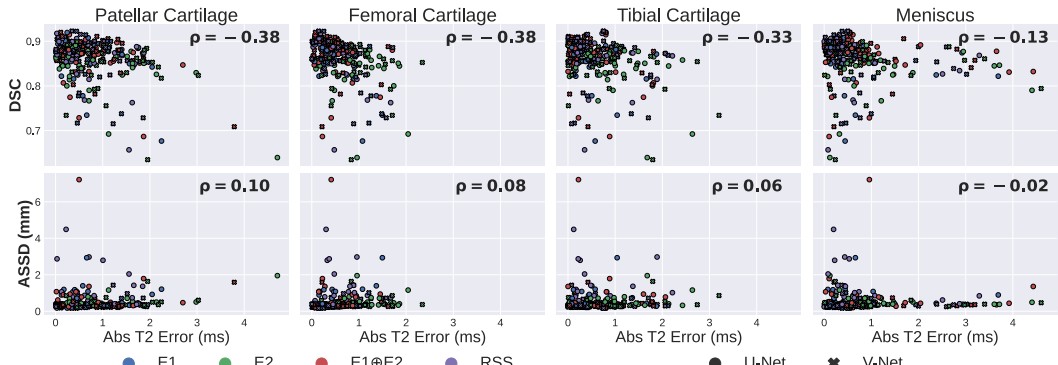

Figure 4: Performance among DICOM Track segmentation models (colored markers) on image quality metrics versus absolute $T_2$ estimation error. Both volumetric (top: dice score coefficient - DSC) and surface distance (bottom: average symmetric surface distance - ASSD) metrics were very weakly correlated ($|\rho| \leq 0.4$) with absolute errors in $T_2$ estimates.

researchers training models on our dataset should account for potential distribution shifts and validate their models on relevant data sampled from their imaging setup prior to deployment. Third, although E2E-VarNet is a popular architecture for MRI reconstruction, it requires training data with a fixed number of coils for learning sensitivity maps [45]. However, scans in SKM-TEA represent real-world data where scans are acquired with different receiver coils with a heterogeneous number of coil elements due to variability in subject sizes. In future work, we will train networks on a subset of scans with the same number of coils. Additionally, there are other anatomical structures in the knee (e.g. bone, muscles), which are not densely annotated, but can help with extracting other clinically-relevant biomarkers. In future work, we will look to curate data from a larger subject pool and multi-vendor scanners and add annotations for these tissues.

## 8 Conclusion

In this work, we introduce SKM-TEA, a quantitative MRI knee dataset that enables clinically-relevant benchmarking of the end-to-end MRI workflow. First, we curate raw data, images, and dense tissue and pathology annotations from 155 clinical MRI scans. Second, we introduce two unique, but complementary, tracks for benchmarking MRI reconstruction, segmentation, and detection methods. Third, we propose an open-source evaluation framework that uses regional qMRI biomarker analysis as a direct endpoint for quantifying model performance. Finally, we train and evaluate state-of-the-art image reconstruction and analysis models on this dataset using the proposed evaluation framework. We find that existing evaluation metrics for both image and segmentation quality are discordant with qMRI biomarkers, particularly in physiologically relevant tissue subregions, which may warrant the need for the proposed evaluation framework for direct estimation of these biomarkers.

By proposing a new dataset, two benchmarking tracks, and an evaluation framework using clinically-relevant qMRI biomarkers in a multi-task manner, we hope that SKM-TEA can enable a wide range of research in methods development and metric design across all stages of the MR imaging pipeline.

## Acknowledgments and Disclosure of Funding

We would like to thank Neel Guha, Beliz Gunel, Megan Leszczynski, Sabri Eyuboglu, Sahaana Suri, and Parth Shah for their feedback on the manuscript.

This work was supported by grants R01 AR077604, R01 EB002524, K24 AR062068, and P41 EB015891 from the NIH; the Precision Health and Integrated Diagnostics Seed Grant from Stanford University; National Science Foundation (DGE 1656518, CCF1763315, CCF1563078); DOD – National Science and Engineering Graduate Fellowship (ARO); Stanford Artificial Intelligence in Medicine and Imaging GCP grant; Stanford Human-Centered Artificial Intelligence GCP grant; GE Healthcare and Philips.

We also gratefully acknowledge the support of NIH under No. U54EB020405 and P41EB027060 (Mobilize), NSF under Nos. CCF1763315 (Beyond Sparsity), CCF1563078 (Volume to Velocity), and 1937301 (RTML); ONR under No. N000141712266 (Unifying Weak Supervision); the Moore Foundation, NXP, Xilinx, LETI-CEA, Intel, IBM, Microsoft, NEC, Toshiba, TSMC, ARM, Hitachi, BASF, Accenture, Ericsson, Qualcomm, Analog Devices, the Okawa Foundation, American Family Insurance, Google Cloud, Salesforce, Total, the Stanford Data Science Initiative (SDSI), and members of the Stanford DAWN project: Facebook, Google, and VMWare.

A. S. Chaudhari has provided consulting services to SkopeMR, Inc., Subtle Medical, Chondrometrics GmbH, Image Analysis Group, Edge Analytics, ICM, and Culvert Engineering; is a shareholder of Subtle Medical, LVIS Corporation, and Brain Key; and receives research support from GE Healthcare and Philips. C.M. Sandino, K.J. Stevens, and G.E. Gold receive research support from GE Healthcare. B. A. Hargreaves is a shareholder of LVIS Corporation; and receives research support from GE Healthcare and Philips. The remaining authors declare that they have no disclosures relevant to the subject matter of this article.

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
