# OpenReview forum: "SKM-TEA: A Dataset for Accelerated MRI Reconstruction with Dense Image Labels for Quantitative Clinical Evaluation"
_NeurIPS.cc/2021/Track/Datasets_and_Benchmarks/Round2 — NeurIPS 2021 Datasets and Benchmarks Track (Round 2)_

### Official Review · Reviewer_6vdh · 2021-09-17
**Knee MRI dataset for end-to-end clinically meaningful AI development**

**Rating:** 7
**Confidence:** 5
**Correctness:** Yes

**Strengths:**

- Comprehensive and meaningful dataset, which helps facilitate the development of AI-based MR reconstruction and quantification of pathologies.
- Well written paper with clear documentation of the dataset description, acquisition protocols, use cases, and example results.
- Bias towards developing quantitative T2 relaxation-based quantification, a clinically relevant biomarker for detecting an early onset of OA.
- Code to replicate the baseline results.


**Weaknesses:**

The authors themselves list the weakness of the paper and do a good job of mentioning how in future work will address each of the weaknesses.
- The limited size of the cohort and its scope limited to just a particular region, hospital, and scanner type.





**Additional Feedback:**

A few minor typos and clarifications:
- Lines 49-50  exists no large-scale public datasets that include both the raw MRI data **along and** multiple clinically- relevant observations for each image.
- Lines 113-114 Classification/Detection: The MRNet and the Osteoarthritis Initiative datasets provide labels for classification of knee MRI abnormalities - Table 1 shows classification crossed out. Please correct the discrepancy.
- Line 205 reconstruction tasks in clinically pertinent regions. When using these labels in the context of **the this**
- Clarification - For the reconstruction track to validate the T2 relaxation, I assume the sub-regions quantification would be based on the ground truth.
- Section 6.4 Raw Data Track – Reconstruction - It's interesting that some of the experiments (Unrolled (E1 E2)) had higher accuracy and lower variance in the 8x  than the 6x. It is surprising that a more aggressive sampling by a factor of 2x produces a higher accuracy. Do the authors have a hypothesis as to why that is happening? It would be a useful addition to the paper.

**Clarity:**

The paper is very well written clearly highlighting all aspects of the data collection usages and its limitations. Further, the suggested use cases and baseline benchmarks and code provide interested users a good starting point to innovate.

**Documentation:**

Yes, the authors provide clear documentation and a plan for how it would be maintained, versioned, and updated going forward.

**Ethics:**

They did get IRB approval and ensured all the PHI information was anonymized. As such I don't have any other concerns.

**Relation To Prior Work:**

The authors clearly compare with the existing datasets for knee MRI and highlight what their differentiation is.

**Summary And Contributions:**

The authors open source a new quantitative MRI dataset along with baseline benchmarks and relevant code. They provided 150 patients with k-space raw data; the scanner reconstructed DICOMs, manually annotated segmentation, and the corresponding bounding boxes. The clear differentiation of this dataset is that it provides for a single patient all the different steps of the MRI imaging modality from acquisition to pathology analysis.

The paper is well written with a clear motivation for the dataset creation, highlighting the different use cases and reporting on a few example baseline models for benchmarking the use cases. As such, I don't have any major concerns about the paper. It reads well. I only list a few minor typos and clarification comments.

---

> ### Author Response · Authors · 2021-09-28
> **Response to Reviewer 6vdh**
>
> We thank the reviewer for their time and invaluable feedback.
>
> **(R4-1)** Grammatical Errors: We thank the reviewer for their attention to detail. We have corrected these errors in the text:
>
> l. 49-50: “...exists no large-scale public datasets that include both the raw MRI data **along with** multiple clinically-relevant observations for each image.”
>
> l. 205: “When using these labels in the context of this track, …”
>
> **(R4-2) MRNet/OAI classification**: We thank the reviewer for raising this point. We have fixed Table 1 to reflect that both MRNet and the OAI dataset have classification labels. We note that the classification labels provided in the OAI dataset are MOAKS scores, which are different from clinical pathology classification. Additionally, the MOAKS scores are only available for specific subsets of scans/patients.
>
> **(R4-3) T2 pipeline clarification**: We thank the reviewer for their note, and they are correct in their assumption. For reconstruction, two T2 maps were computed, one from the DL-reconstructed images and one from the ground truth SENSE images. The gradient-warp-corrected segmentations were used as ground truth segmentations to extract sub-regions estimates from each of the T2 maps.
> We have provided these details in Appendix C and have referenced this appendix in Section 6.1.
>
> **(R4-4) 6x vs 8x Reconstruction Performance**: We thank the reviewer for this point. For articular (patellar, femoral, tibial) cartilage, E1$\oplus$E2 networks have similar or higher accuracy and lower variance at 8x compared to 6x. However, for other networks (E1/E2, E1+E2), this was not observed. Because E1 has more signal in articular cartilage compared to E2, joint optimization of multi-channel E1-E2 inputs may result in over-prioritization of reconstruction of the E1. This may lead to more variable per-pixel recovery for E2. As a result, per-pixel T2 maps, which are computed based on pixel-wise ratios between E2/E1, may have more variability and error. At higher acceleration factors, networks optimized with the l1-loss tend to blur the image (i.e. average the signal), and can reduce the per-pixel variance and lead to more accurate region-wide signal, and thus T2, estimates.
>
> For reconstruction, we also note that all networks at 8x underperform corresponding networks at 6x among conventional image quality metrics (pSNR, SSIM). This observation may reaffirm that existing image quality metrics are not sufficient predictors of clinically relevant metrics like quantitative MRI parameter estimates.
>
> We have added this discussion to Section 6.4.
>
> **(R4-5)** We thank the reviewer for their invaluable feedback and time. We have addressed all points brought up by the reviewer: grammatical errors, dataset table correction, T2 pipeline clarification, and 6x vs 8x T2 accuracy/variability comparison. We hope that these responses are deemed satisfactory to the reviewer.

---

### Official Review · Reviewer_CV6s · 2021-09-19
**Good dataset covering data from k-space to quantitative biomarkers. Has potential.**

**Rating:** 6
**Confidence:** 3
**Clarity:** Yes, the paper is very well-written.

**Strengths:**

1. The multi-task nature of the dataset. As highlighted in the paper, presence of k-space data along with clinical labels has the potential to open up numerous interesting research questions.

2. The care given to differentiating scanner-based reconstructions (using proprietary software) and the SENSE reconstructions, and cautioning the reader on their corresponding usage is admirable.

3. The paper is very well written and easy to follow.

**Weaknesses:**

1. The size of the dataset (155 patients), with scans from a single centre and manufacturer still makes trustworthy clinical translation difficult. To be fair, the authors acknowledge this in the conclusions. However, a more solid claim is needed to ensure that the dataset will actually evolve over time, given its potential.

2. The benchmarking over the subset of tasks seems rushed. Why were only a subset of tasks benchmarked? Benchmarking classification/anomaly-detection would have been a more clinical relevant step. Also, there exist cutting edge techniques for k-space to image reconstruction. An explanation is warranted for why only U-Net and unrolled networks were used. Also, no intuitions or learnings are provided based on the observed performances.

3. Information on annotation procedure and inter-rater variability is missing. This could answer the question: How reliable are the segmentation masks or the bounding boxes or even the diagnoses?

**Additional Feedback:**

In my opinion, the work has immense potential. In its current form, even though the dataset is complete, the benchmarking seems incomplete. The latter not being the primary focus of the dataset track is why I'm recommending an 'accept'. However, a complete benchmark plus the dataset would make this work very strong.

**Correctness:**

Yes, the dataset is constructed properly. Why was this split chosen? Was it random? Based on the provided table in the supplement and the well-written text, it is clear to deduce the experimental design. However, an improved version of Table 6 (supplementary) mentioning the actual 'Tasks' for which the dataset can be used (incl. but not exclusively, of course) would be very helpful.

**Documentation:**

Yes, relevant documentation is provided. Training details of the benchmarking process have been provided. But, actual implementation is missing, whose presence could strengthen the reproducibility manifold.

**Ethics:**

As data is de-identified and reviewed by the IRB, I don't readily see any concerns.

**Relation To Prior Work:**

Table 1 and Section 3 do the job. Why not have actual dataset sizes in Table 1? Is it hard to determine for the other datasets?

**Summary And Contributions:**

This paper provides a multi-task knee MRI dataset of 155 patients. This includes the 'full resolution' k-space data, the scanner and SENSE reconstructions, the quantitative (T2) maps, as well as dense segmentations of some anatomies as well as bounding box annotations of some anomalies. Therefore, the paper boasts of a wide gamut of data types from different parts of the MR reconstruction and analysis pipeline. Additionally, thanks to the availability of the parameter maps, this work proposes an evaluation protocol closer to clinical usage, instead of the traditional reconstruction and analysis measures.

---

> ### Author Response · Authors · 2021-09-28
> **Response to Reviewer CV6s**
>
> We thank the reviewer for their time and their insightful remarks. We have added clarifications to each of the comments below. We have also added responses in a following comment due to character limits.
>
> **(R3-1) Dataset Size and Multi-Site/Multi-Vendor Extensions**: We thank the reviewer for this note. In Section 7, we note that we plan to extend our data collection to different sites, which include different vendors: “In future work, we will look to curate data from a larger subject pool and multi-vendor scanners and add annotations for these tissues.” Additionally, the semantic versioning of the dataset and annotations (see Appendix A.4) will allow future users to track the growth and evolution of this dataset.
>
> **(R3-2) Benchmarking for SKM-TEA**
> - *Benchmarking detection*: The reviewer raises a great point about the benchmarking of classification/anomaly-detection algorithms. We plan to keep the benchmarking process “alive” and will be continuously adding benchmarks as the dataset evolves. We are currently exploring training a Mask R-CNN network for baseline detection performance.
> - *Cutting edge techniques for reconstruction*: Among DL approaches, unrolled and fully convolutional networks have achieved the highest performance [1]. Compressed sensing reconstruction requires careful and extensive fine-tuning of regularization parameters per slice, and thus can be intractable to large datasets like SKM-TEA. However, for completeness, we will add benchmarks for E2E Var-Net [2], a reconstruction model that has shown success on the fastMRI dataset.
> - *Insights from benchmarking*: We have included the following discussion points around observations in model performance in Section 6.4
> Unrolled networks outperformed U-net networks for reconstruction. In standard fully convolutional networks (FCNs) like U-Net, the network can only leverage statistical imaging priors learned during training. However, unrolled networks pair these imaging priors with proximal update steps that can encode priors about the image formation forward model, which can improve image recovery.
> Segmentation networks trained only on E2 perform worse for T2 estimation than E1/RSS/E1$\oplus$E2 networks. The second echo (E2) of the qDESS sequence has considerably lower signal (and thus SNR) compared to the first echo (E1). This may result in poorer segmentation performance among networks trained only with E2, and thus higher error in T2 estimates.
>
> [1] Hammernik, K., Schlemper, J., Qin, C., Duan, J., Summers, R. M., & Rueckert, D. (2021). Systematic evaluation of iterative deep neural networks for fast parallel MRI reconstruction with sensitivity‐weighted coil combination. Magnetic Resonance in Medicine.
>
> [2] Sriram, A., Zbontar, J., Murrell, T., Defazio, A., Zitnick, C. L., Yakubova, N., ... & Johnson, P. (2020, October). End-to-end variational networks for accelerated MRI reconstruction. In International Conference on Medical Image Computing and Computer-Assisted Intervention (pp. 64-73). Springer, Cham

---

> > ### Author Response · Authors · 2021-09-28
> > **Response to Reviewer CV6s contd**
> >
> >
> > **(R3-3) Annotation Procedure/Inter-reader Variability**: We thank the reviewer for this note. Tissue segmentations and detection bounding boxes were created by two researchers with 4 and 3 years experience with knee MR image interpretation, under the auspices of two board-certified musculoskeletal radiologists with 26 and 24 years of experience. Labels were collected disjointly between both annotators, such that no one scan had labels from multiple annotators. As such there were no disagreements between annotators for the same scan. In future updates to this dataset, we will consider adding new readers to annotate the test set to provide intra-reader variability. We have included details for the annotation procedure in Appendix A.4.
> >
> > The reviewer also raises a good point about inter-reader variability. We discuss these points for the different annotations below. In future updates to this dataset, we will consider adding new readers to annotate the test set to provide intra-reader variability.
> > - Segmentation labels: Given the density of segmentation labels, we elected to collect labels for a larger number of scans to construct a larger dataset in place of multiple segmentations for the same scan. As such, we cannot compute the inter-reader variability for these cases. However, we note that the exact MRI sequence used in this study has shown very high scan-rescan human precision with segmentation [1].
> > - Detection labels: Though we do not have access to inter-reader variability measures for bounding box labels, we note that even experienced radiologists have discordance in determining the extent of an abnormality. However, from a clinical perspective, localizing the abnormality in the image can be beneficial, compared to a single classification label that may exist anywhere in a 3D image.
> > - Diagnosis labels: High-quality diagnoses were obtained from clinical radiology reports created by board-certified radiologists that were used for patient care. These reports were subsequently transcribed into bounding boxes by the annotators under the supervision of two board-certified musculoskeletal radiologists with 26 and 24 years of experience.
> >
> > [1] Chaudhari, A. S., Black, M. S., Eijgenraam, S., Wirth, W., Maschek, S., Sveinsson, B., ... & Hargreaves, B. A. (2018). Five‐minute knee MRI for simultaneous morphometry and T2 relaxometry of cartilage and meniscus and for semiquantitative radiological assessment using double‐echo in steady‐state at 3T. Journal of Magnetic Resonance Imaging, 47(5), 1328-1341.
> >
> > **(R3-4) Dataset Split and Task Details**: We thank the reviewer for their comment. The test split was curated from 36 patients who received additional arthroscopic surgical intervention (see Section 4.2). This test set was chosen for ​​a subset of patients that underwent arthroscopic surgery based on the results of their original MRI scan. As such, we could confirm the presence of the injuries noted during imaging since they were apparent during the surgery. Thus, the clinical annotations in the test set are denser and concordant with arthroscopic findings. The remaining scans were separated into training and validation splits randomly. We have added these details to Section 4.2: Data Preparation.
> >
> > **(R3-5) Inclusion of Dataset Sizes**: We thank the reviewer for this note. The dataset sizes are difficult to exactly quantify because of differences in how size quantification is done in practice. Volumetric medical imaging like MRI is unique in that the extent of a dataset can be quantified in multiple ways - number of scans, number of patients, number of slices, number of longitudinal scans, etc. Even among slices, there are differences along which plane to count slices. To avoid overspecifying dataset sizes based on explicit slice counts, we provide a more high-level categorization of size, which still communicates the general magnitude of different datasets.
> >
> > **(R3-6) Code Availability**: We thank the reviewer for this important point and strongly agree that the availability of implementation is crucial. The implementation is currently provided in the supplementary material but will be open-sourced with available tutorials upon acceptance.
> >
> > **(R3-7)** We thank the reviewer for their invaluable feedback and time. We have addressed all major points brought up by the reviewer: additional benchmarks, annotation procedure, inter-reader variability, dataset sizes, and code availability. We hope that these responses are deemed satisfactory to the reviewer. We will continue benchmarking more segmentation, reconstruction, and detection models on this dataset.

---

### Official Review · Reviewer_k4UR · 2021-09-20
**Review of SKM-TEA**

**Rating:** 9
**Confidence:** 4
**Clarity:** The paper was very well written, clea…

**Strengths:**

1. The dataset provides a large subset of patients that can be very useful in clinical research and is useful for various clinical research areas (segmentation, reconstruction, classification, detection).
2. An extensive evaluation of the dataset was carried out by the authors and the results are clearly analyzed.


**Weaknesses:**

1. The data is missing information on the annotators:
    a) How the annotators were selected
    b) The ground truth criteria the annotators used
    c) How disagreements in annotations were resolved.
2. The authors also failed to provide Information on how the dataset will be accessed in future.


**Additional Feedback:**

Overall this is a very well written paper. the motivation for the paper is clear and the reviewer believes it is a significant contribution, however, the reviewer has a few concerns on the annotators of the dataset (i.e. how the annotators were selected, The ground truth criteria the annotators used, and how disagreements in annotations were resolved). Furthermore, more information is needed on how the dataset will be accessed in the future, and the plans for maintaining the dataset.

EDIT. Thank you for the response, the review has been updated

**Correctness:**

Although the paper was very well written and the motivation clearly described in the paper, The reviewer failed to see a detailed description on how the dataset was constructed. Therefore it is difficult to determine the overall accuracy of the dataset.

**Documentation:**

There is a adequate information on the data collection and organization as well as ethical and responsible use. however, information on the availability and maintenance of the dataset is not clearly stated.

**Relation To Prior Work:**

The authors clearly described previous works and how their work differs from them

**Summary And Contributions:**

The authors propose a new dataset for quantitative MRI scans, the corresponding scanner-generated DICOM images, segmentations of tissues, and bounding box annotations for sixteen clinically relevant pathologies. They also provide a benchmark evaluation method for an accurate and consistent metric to measure tissue health. finally they extensively test the dataset for various tasks.

---

> ### Author Response · Authors · 2021-09-28
> **Response to Reviewer k4UR**
>
> We thank the reviewer for their time and useful comments. We have added clarifications to each of the comments below
>
> **(R2-1) Annotator and annotation information**: We thank the reviewer for their insightful notes. We have summarized the details below and included these details in Appendix A.4.
>
> a,b) Tissue segmentations and detection bounding boxes were created by two researchers with 4 and 3 years experience with knee MR image interpretation, under the auspices of two board-certified musculoskeletal radiologists with 26 and 24 years of experience. All four individuals had semi-structured clinical radiology reports available for bounding box creation. The start and end slice of the images in all three image orientations (sagittal, axial, coronal) where the abnormality noted in the report was visible was used to create 3D bounding boxes. We note that even experienced radiologists have discordance in determining the extent of an abnormality, but from a clinical perspective, simply localizing the abnormality in the image can be beneficial, compared to a single classification label that may exist anywhere in a 3D image.
>
> For cartilage and meniscus segmentation, annotators used both qDESS echoes that provide separate image contrasts to distinguish between the neighboring cartilage and meniscus pixels, as well as additional tissues such as bone, muscle, and joint fluid. Segmentations were performed slice-by-slice in the sagittal plane and volumetric consistency was enforced by correcting segmentations in the axial and coronal planes in the ITK-SNAP software. Every image volume segmentation was quality controlled by the two researchers with 4 and 3 years experience with knee MR image interpretation
>
> c) Labels were collected disjointly between both annotators, such that no one scan had labels from multiple annotators. As such there were no disagreements between annotators for the same scan. In future updates to this dataset, we will consider adding new readers to annotate the test set to provide intra-reader variability.
>
> **(R2-2) Dataset Availability and Maintenance**: We thank the reviewer for this note. Data will be hosted and maintained by the authors and Microsoft Azure as part of a partnership with the Stanford Center for Artificial Intelligence in Medicine and Imaging (AIMI). Regarding maintenance, all data and corresponding artifacts (annotations, etc.) will be semantically versioned and available for future use. The dataset will be maintained by the authors in cooperation with the Stanford AIMI Center. A copy will also be stored on Google Drive for redundancy. Details on how the dataset will be distributed and maintained are provided in Appendix A.4. We have added a reference to this appendix in Section 4.2 where the distribution of the dataset is discussed.
>
> **(R2-3) Dataset Construction**: We thank the reviewer for this point. Section 4 in the main text and Appendix A provide details for data collection, preprocessing, distribution, and maintenance. Briefly, consecutive patients at Stanford Hospital who were scheduled to receive a diagnostic knee MRI scan were included in this study, if they consented. As the reviewer mentioned in R2-1, we have also added a description of the annotation process in Appendix A.4.
>
> **(R2-4)**: We thank the reviewer for their feedback. Overall, we have addressed all major points brought up during the initial review: annotator selection, ground-truth criteria for image annotation, annotator disagreements, dataset construction, and dataset construction. We hope that these responses are deemed satisfactory to the reviewer.

---

### Official Review · Reviewer_X3o1 · 2021-09-22

**Rating:** 6
**Confidence:** 3
**Clarity:** The paper is well written.

**Strengths:**

* The paper is well-motivated with clinical contributions. The collection of large-scale labelled knee segmentation and reconstruction data is difficult and as a result benchmarking would benefit corresponding clinical applications.

* The authors divide their evaluation methods into Raw Data Benchmarking Track and the DICOM Benchmarking track.

* The data collection and pre-processing is explained.

* The authors report accuracies over different knee segments which is helpful in potentially understanding of potential diseases.

**Weaknesses:**

The model is used is U-Net which has proven to show great results for segmentations and reconstruction. Currently there are better models nn-Unet and residual based U-Net and analyses on such models is missing. While the dataset produced is large, its functionality remains undetermined if results are not compared with those of other knee datasets.

**Additional Feedback:**

It would be better to provide comparison with existing benchmarks and newer models.

**Correctness:**

Yes, the claims made in the submission are correct. The evaluation methods and experiment design are appropriate.

**Documentation:**

Yes, sufficient detail is provided. The authors must create a tutorial .ipnyb or README for reproducibility.

**Ethics:**

The author discussed ethical issues in the paper therefore none.

**Relation To Prior Work:**

Prior works have been well discussed.

**Summary And Contributions:**

The paper presents a dataset of quantitative MRI with knee scans, images, and dense labels to create a localized tissue-wise qMRI biomarker maps benchmarking of the MRI reconstruction. The paper benchmarks MRI reconstruction and segmentation models using both traditional and new deep learning image and label metrics.

---

> ### Author Response · Authors · 2021-09-28
> **Response to Reviewer X3o1**
>
> We thank the reviewer for their time and their invaluable feedback. We have added responses to each of the suggestions below
>
> **(R1-1) Benchmarking newer models besides U-Net and Unrolled**: We thank the reviewer for this point. Recent results from challenges and standardized studies for both MR knee segmentation [1] and reconstruction [2] have found that the differences in performance of similar families of network architectures (e.g. fully convolutional networks) are minor. The difference between accuracy (SSIM for reconstruction, Dice for segmentation) among best and worst performing models for both tasks in these studies was ~2%. In the case of reconstruction, the family of the network architecture -- fully-convolutional (i.e U-Net, V-Net, etc.) vs unrolled has a larger impact on performance, rather than how those networks are specifically implemented. As a result, for the reconstruction task, we selected a sample architecture from each family for comparison.
>
> The reviewer raises a great point that having these models benchmarked would be useful. We will include segmentation results from both the residual U-Net and the V-Net, another residual fully convolutional network, in the final version. However, due to model training time limitations and the need for careful hyperparameter optimization, we cannot include those results currently. We will add the results of these models to the final version of the paper upon availability.
>
> [1] Desai, A. D., Caliva, F., Iriondo, C., Mortazi, A., Jambawalikar, S., Bagci, U., ... & IWOAI Segmentation Challenge Writing Group. (2021). The international workshop on osteoarthritis imaging knee MRI segmentation challenge: a multi-institute evaluation and analysis framework on a standardized dataset. Radiology: Artificial Intelligence, 3(3), e200078.
>
> [2] Hammernik, K., Schlemper, J., Qin, C., Duan, J., Summers, R. M., & Rueckert, D. (2021). Systematic evaluation of iterative deep neural networks for fast parallel MRI reconstruction with sensitivity‐weighted coil combination. Magnetic Resonance in Medicine.
>
> **(R1-2) Comparison to other datasets**: We thank the reviewer for this comment. We acknowledge that understanding the dynamics between different datasets would help provide a broader understanding of these datasets. However, strict inter-dataset comparison may be difficult due to critical differences in both dataset properties (e.g. acquisition type, 2D vs 3D MRI, MRI scanner parameters), processing techniques (e.g. normalization, metric computation), and labels available (reconstruction, segmentation, detection, classification). Though these factors may make comparing results between different datasets difficult, the functionality of the dataset may largely be assessed based on the utility that the acquired images have provided in the past for clinical and research studies (see references [12,13,15] in manuscript).
>
> **(R1-3) Tutorial creation**: We strongly agree with the reviewer. The current code submitted as supplementary material contains a python notebook (train.ipynb) with step-by-step instructions to familiarize users with the codebase and experiments for this particular submission. We also provide a README (https://github.com/StanfordMIMI/skm-tea) that details the structure of the dataset and annotations. We will provide clear references to both in the final code release.
>
> **(R1-4)** We thank the reviewer for their time and feedback for this manuscript. We have addressed all major points brought up during the initial review: benchmarking of newer models, comparison to other datasets, and tutorial creation. We hope that these responses are deemed satisfactory to the reviewer.

---

### Author Response · Authors · 2021-09-28
**Response and Revision Summary**

We thank the reviewers for their thoughtful reviews and insightful comments. We are glad that they appreciated our work’s contribution in curating a new dataset to enable multiple relevant medical imaging tasks. The reviewers raised some excellent questions, and we have addressed each suggestion in detail in their respective threads. A summary of all major changes to the manuscript is also indicated below.

1. Annotator details were added in Appendix A.4
2. Details regarding how T2 error was computed were added in Appendix C
3. Takeaways from observations were added to section 6.4.

---

### Decision · Program_Chairs · 2021-10-09

**Decision:**

Accept

**Comment:**

The reviewers all liked the paper. The authors' response clarified some important points. In view of that, the authors are strongly invited to take the feedback on board for the final version.